# Registered Nurses’ Views and Experiences of Delivering Care in War and Conflict Areas: A Systematic Review

**DOI:** 10.3390/healthcare10112168

**Published:** 2022-10-29

**Authors:** Abdullah Sadhaan, Michael Brown, Derek McLaughlin

**Affiliations:** 1School of Nursing and Midwifery, Queen’s University Belfast, Belfast BT97BL, UK; 2Trauma Department, King Saud Medical City, Riyadh 12746, Saudi Arabia; 3Mental Health Nursing, Queen’s University Belfast, Belfast BT97BL, UK

**Keywords:** experience, military nursing, nurse, systematic review, war and conflict area

## Abstract

Background: Several studies have been undertaken regarding civilian and military nurses’ training, deployment, and experiences during missions in war and conflict areas. However, no review study regarding the experiences of nurses in serving in war and conflict areas has been published. Aim: This review aims to identify the views, experiences, and support needs of Registered Nurses when caring for patients in war and conflict areas. Method: Four electronic databases—MEDLINE, CINAHL, PsycINFO, and general BC PubMed—were searched in this systematic review. Study screening and selection, data extraction, quality appraisal, and narrative synthesis were conducted following the Preferred Reporting Items for Systematic Reviews and Meta-Analyses (PRISMA) 2020 checklist. Results: Twenty-five studies were included in the final review. The findings were categorised based on four main themes: Challenges in nursing practice, Meaning of experience, Scope of practice, and Nursing support pre- and post-conflict. Conclusions: Registered Nurses play a critical role in prehospital care and have a significant impact on the survival of wounded military personnel and civilians and on their mortality. Registered Nurses play important roles in military deployment, with barriers to their successful implementation experienced due to a lack of information at the decision-making level, and the need for psychological supports and role-appropriate medical readiness training. Implications: Registered Nurses who are military-ready need to be effective in war and conflict areas. Using the experiences of military or civilian Registered Nurses to assess the pre-deployment preparation needs of these nurses will be beneficial to the effectiveness of the nursing workforce. There are elements that demand more attention during the pre-deployment preparation phase of nurses required to serve in war and conflict areas.

## 1. Introduction

During the Crimean War in 1854, Florence Nightingale established herself as the mother of modern nursing, where she led a group of volunteer nurses on a mission to help injured soldiers [1]. Another example is the nurses who volunteered for both the union and the confederate forces during the Civil War in the United States of America (USA), who used many of Nightingale’s principles and nursing practices [2,3]. During the Spanish–American War in 1898, the military hired contract female doctors and nurses [4]. Over time, an increasing number of nurses have developed knowledge and skills in this area. More recently, about 59,000 civilian nurses served in the armed forces during World War II to care for troops, family members, and civilians [5]. Therefore, nurses have a long history of providing care to military personnel and civilians in war and conflict areas.

By definition, war is an “open and declared conflict between the armed forces of two or more states or nations” [6], whereas conflict is described as “a competition of political and human will that can use violent and non-violent means to influence a diverse group of actors to achieve the political objective” [7]. However, not all conflict is termed war, having a spectrum that extends from no-conflict situations (e.g., humanitarian relief) up to and involving total war between nations or states [7].

## 2. Background

Registered Nurses are critical for infectious disease detection and prevention, social care, and recovery during terrorism and war and conflicts [8,9]. Civilian Registered Nurses provide much of the treatment in these situations as they are readily available, have essential clinical and organisational skills, and are generally trusted and respected by local populations [9,10]. However, the area of nursing practice has room to grow in terms of systems training, capacity growth, and engagement in policy and analysis related to war and conflict areas. Civilian Registered Nurses, on the other hand appear to lack specific training to prepare them for roles in war and conflict areas [11,12]. This is due in part to unresolved ethical and political concerns among nursing leaders regarding the profession’s reputation and role, humanist ideals, and relationships with the government [11,12].

The care and support provided by Registered Nurses serving in the armed forces have some similarities to civilian nursing, with the main difference being that Registered Nurses specialising in providing care to patients in war and conflict areas across the world [13]. Registered Nurses in the armed forces practice in the army, navy, and air force, with some also working in the fields of education and administration. They provide nursing care to military personnel when on active duty in war and conflict areas and personnel’s families at home. These Registered Nurses provide care and support for injured military personnel, and care for civilians’ casualties resulting from, for example conflicts or natural disasters [14,15]. When there is war or conflict, both armed forces and civilian Registered Nurses undertake a range of roles, such as triage. Triage is the process of assigning degrees of urgency to the management of wounds or conditions to prioritise care based on immediate assessed clinical needs [15,16,17]. Both armed forces and civilian Registered Nurses deliver a variety of roles outside of war, providing care and support for current and former armed forces personnel and their families [14,16]. Military Registered Nurses serve at sea in war and conflict areas, and on military bases, which can be fast-paced, stressful, and uncertain [9,10,14,16]. Consequently, Registered Nurses provide care and support to patients who have experienced significant biological and psychological issues, including the possible consequences on their military career as a result of injuries and trauma [9,10,14,16].

At established armed forces treatment facilities, similar to civilian hospital systems, Registered Nurses develop key nursing skills and critical thinking abilities to enable the provision of care and support [18]. Registered Nurses in the armed forces must also be aware of best practices in civilian hospital settings to enable the provision of care and support in situations that are unregulated and with limited resources to deliver safe and effective care where normal treatment cannot be provided or is impractical [8,18]. According to Tom-James [8], a clear understanding of the evidence that informs nursing practice equips Registered Nurses in the armed forces with the expertise and skills to adapt nursing care appropriately, for example during transport, under fire, or on rough seas [8].

While research has been undertaken regarding the training, deployment, and experiences of Registered Nurses in war and conflict areas, no review study of the published studies has been conducted regarding civilian and military nurses’ experience serving in war and conflict areas. Therefore, the aim of this review is to identify the views and experiences and support needs of civilian and military Registered Nurses when providing care and support for patients in war and conflict areas.

## 3. Materials and Methods

### 3.1. Search Strategy

This systematic review adhered to the Preferred Reporting Items for Systematic Reviews and Meta-Analyses (PRISMA) 2020 checklist [19]. The following electronic databases: MEDLINE, CINAHL, PsycINFO, and general BC PubMed were searched from their inception dates until February 2022, using MeSH keywords and Boolean operators. The researchers chose the most commonly used database (i.e., MEDLINE) and other three databases to capture studies relevant to the eligibility criteria of this review. With the help of a subject librarian, the following key terms were used: trauma, multiple physical trauma, damage, injury, wound, traumatization, collapse, nurs*, nursing, nursing care, healthcare provider, staff, trauma nurs*, critical nurs*, health professional, experiences, view, understanding, perspective, conflict and war.

### 3.2. Eligibility Criteria

The inclusion criteria used were full-text quantitative studies, qualitative studies, or mixed method studies written in English language and conducted in any country. Opinion articles, editorials, commentaries, studies involving other health professionals and not involving Registered Nurses, and studies not published in English language were excluded.

### 3.3. Study Screening

The researchers used Covidence Systematic Review Software 2021 for importing references from databases used in the searches. Based on the eligibility criteria, two researchers independently assessed the studies by titles and abstract. Then, two researchers (A.S. & M.B.) independently reviewed each full text of the eligible studies, with conflicts resolved by a third researcher (D.M.). The PRISMA 2020 statement guided and enabled the researchers to report this systematic review that allows the readership to evaluate the appropriateness of the methods used, leading to the trustworthiness of the review findings. The PRISMA flow chart of this process including searching, screening and selecting the studies is detailed in Figure 1.

### 3.4. Data Extraction Process

A data extraction form was developed by the researchers to facilitate the process. The primary researcher (A.S.) independently extracted data on year, country, authors, study aim, data collection methods, key findings and research recommendations regarding the experiences of nurses in war and conflict areas. Separately, the results of the identified studies were critically reviewed and summarised in Table 1. Endnote was used to store the information extracted because it is ideal for recording the process and making reference lists for the study easier [20].

### 3.5. Quality Appraisal and Study Synthesis

Each article was critically appraised using the Mixed Methods Appraisal Tool (MMAT), designed to evaluate mixed studies that includes qualitative, quantitative, and mixed-method designs [21]. Since the included studies are heterogeneous, this review utilised a narrative synthesis of the findings based on Popay et al.’s [22] guidelines. The researchers adhered to the following three iterative steps: (i) studies organised into logical categories, (ii) studies compared and findings synthesised, and (iii) analyse of findings within each theme to identify interconnectedness within and between the studies.

## 4. Results

### 4.1. Search Outcomes

A total of 5148 papers were initially identified across the databases from the search process, leaving 3551 after duplicates were removed. The search was narrowed down by objectively evaluating the selected studies considering the aim and primary question of the review. Based on the eligibility criteria, titles (*n* = 2429) and abstracts (*n* = 1001) were excluded. Of this, 121 studies remained for full-text review, with a further 96 studies excluded after full-text screening. A total of 25 full-text articles, 23 qualitative studies, one quantitative study and one mixed-method study, were included in the final review.

### 4.2. Characteristics of Included Studies

Of the 25 included studies, three studies used questionnaire in data collection [23,24,25], and four studies used focus group interviews [26,27,28,29]. The 18 remaining studies used face to face semi-structured interviews with an average sample size of *n* = 14 (Range = 4–37).

Regarding to the country where the studies were conducted, five studies undertaken in the USA [26,30,31,32,33]. Two studies were conducted in Australia [34,35] and two in Iran [36,37]. One study was conducted in the USA and Europe [23], New Zealand [38], Palestine [24], Norway [25], and Sweden [39].

Five studies achieved a score of 100% using the MMAT quality appraisal tool 100% [27,30,33,39,40]. Twelve studies had a quality score of 75% [23,29,34,35,36,38,41,42,43,44,45]. Five studies achieved an MMAT score of 50% [24,28,33,35,46] and the remaining three studies a 25% score [25,26,31]. Out of the 25 studies, 23 studies that had clear research questions, while the two remaining studies the research question could not be identified [31,41]. One study was unclear if the qualitative data collection methods were adequate to address the research question [41]. Four qualitative studies were assessed as inadequate to address the research question [28,33,44,46]. Five studies lacked details of the interpretation of results based on the analysis of the data [23,29,35,38,46]. Five studies lacked coherence between qualitative data sources, collection, analysis, and interpretation [31,34,36,37,43].

### 4.3. Narrative Synthesis of Results

As included studies were heterogeneous, therefore, this review utilised a narrative synthesis approach [22]. A narrative synthesis is defined as “an approach to the systematic review and synthesis of findings from multiple studies that relies primarily on the use of words and text to summarize and explain the findings of the synthesis” [22]. The narrative synthesis of this review resulted in four themes: (i) challenges in nursing practice, (ii) meaning of experience, (iii) scope of practice, and (iv) nursing support pre- and post-war and conflict.

#### 4.3.1. Challenges in Nursing Practice

The nurses’ lived experience was one of personal and professional challenge and growth, with both sad and rewarding clinical situations, physical danger and tiredness, and professional and personal uncertainty in an unstable, and lack-of-privacy setting [26,41]. The Registered Nurses struggled with the moral and ethical challenges that come with working in war and conflict areas, and those associated being outnumbered by members of the armed forces, the distant from home, and stationed in a new country. Some study participants cited physical problems related to training and equipment. Scannell-Desch and Doherty [45] highlighted the essence of this lower capacity for caring, which is like the findings of the studies. Further research was recommended to identify strategies that enabled civilian and military Registered Nurses to avoid compassion fatigue and guilt following deployment.

To practice effectively, Registered Nurses in the armed forces required to have the necessary clinical competences and skills, with the assessment and management of pain acknowledged as being of high quality and responsive to patient needs [25,26,27,38,45]. However, not all of care was of the high quality for a variety of reasons, including inattention to patient needs; leaving lights on late at night; making too much noise when patients were trying to sleep; and gaps in care or administration, such as medications not being available during patient transfers, all contributing to discomfort and frustrations. Some Registered Nurses expressed a lack of confidence in their ability to address these flaws in others’ behaviour without provoking personal criticism [31,38].

Returning home after deployment was described as stressful and difficult to rationalise, even though they were now at home [23,32,40,45,46]. Participants discussed the difficulties of figuring out where they fit in a noncombat environment. Wounded service veterans or retired military Registered Nurses reflected on their pre-war and post-war selves as they struggled with the challenges of building a post-war identity [24,38]. They were not the same person anymore, and the qualities required for future directions were unclear. They were aware of how the outside world saw them and how this influenced how they defined themselves. The situational pressures connected with a deployment might shift quickly because of the metamorphosis and severe dynamic flux that were frequently observed in war and conflict areas [36]. It was important to be able to communicate with home readily and on a frequent basis [36]. In addition, there were interventions identified that played a significant role in reducing psychological stress at war and conflict areas and were well received when properly implemented, such as training, working under a solid command structure, the need for rest, recuperation, exercise, and food [40]. Despite this, there was no formal debriefing model available for those who needed to discuss traumatic occurrences such as professional supervision [40]. Consequently, some study participants reported feeling isolated and unable to express their problems [40,45].

#### 4.3.2. Meaning of Experience

People change as individuals because of their experiences in war and conflicts, as they forfeit what and who they were. Some study participants reported struggles to adjust to life at home, as well as a strengthening family bonds [30,40,46]. Some of the adjustments were viewed positively by some study participants, discovering they had grown stronger, with organisational and leadership abilities were learned, developed, and enhanced. In a Swedish one study, participants reported their personal growth after serving in the armed forces [45].

Registered Nurses in the armed forces were referred to as ‘warriors’ with a dual professional position, both compatible with the professions’ codes and principles, thereby enabling the focus on the operational role [42]. An important element of this was the process of socialisation, training, and identification of the two roles, thereby enabling integration within the larger armed forces system. In one study participants were of the view that their preparations were sufficient [28]. They pondered on their previous experiences within clinical areas such as emergency care, thereby leading to a greater feeling of being preparedness for the roles ahead. Some participants extended their roles in the armed forces by taking on additional responsibilities despite potential barriers such as different religious and cultural values, limited resources, and linguistic constraints [25,32,45]. These roles involved increased clinical decision-making, teaching and mentorship roles, as well as leadership and management capabilities despite the discomfort that may come when working on the fringes of professional boundaries. Consequently, many study participants discovered within themselves the ability to apply innovation in practice [25,32,45].

#### 4.3.3. Scope of Practice

Central to the professional practice of Registered Nurses, referral care was created, featuring first line, second line, and third line levels of care, where they provided expert assessment, treatment and care for casualties during a conflict [36]. For many study participants, task management, triage, and prioritisation, and practice innovations, were among the accomplishments of working throughout conflicts and learning from them. Practice innovations resulting in new approaches and solutions in patient care [27]. Participants in some studies saw in themselves the potential to contribute creative ideas and saw the conflict as an opportunity to develop and share new practice and new ways of working that improved care and support [23,27,29,31,43]. Others highlighted that it was crucial to have the right mindset to give high-quality care, despite being called upon to compensate for lack of equipment and supplies that would be found in fixed health facilities and were not always available in the field [37,44,46].

#### 4.3.4. Nursing Support Pre- and Post-War and Conflict

Before being sent in a combat area, participants detailed how they required to be clinically equipped with core medical-surgical and trauma nursing skills [26]. To support this, nurse education programs with a focus on psychological resilience, emotional well-being, and the consequences of experiencing extreme events were required [24,40,44]. This was seen as necessary to address memories of events that were painful and disturbing for years after they had occurred [24,40,44]. The reintegration process post-conflict back into civilian life was characterised by briefings that were often helpful and supportive to participants. It was highlighted in some studies that the reintegration process was at times facilitated by trainers who appeared indifferent and focused on getting the briefings over as quickly as possible. However, the necessity of time to enable the reintegration process was stressed by some study participants [29,38,46].

## 5. Discussion

This review identified that Registered Nurses who experienced delivering care to wounded armed forces and civilian patients during war and conflict areas struggled. Preparation before deployment and substantial support post-deployment are vital elements for the success of the roles of Registered Nurses in war and conflict areas. Specifically, Registered Nurses in the armed forces needed to have adequate preparation and training for diverse situations that they are required to respond to in times of danger and uncertainty positions in war and conflict areas [28]. In contrast, civilian Registered Nurses worked and provided care to patients without specific preparation and training to prepare them for roles in war and/or conflict areas [11,12]. Despite the differences that existed between civilian and Registered Nurses in the armed forces, this review found evidence that both were affected with many challenges and struggles such as, emotional, ethical/moral, mental and physical issues before and even more after deployment to war zones and/or conflict areas [24,26,27,29,30,32,35,37,40,41,42,44].

The evidence of the experience of both armed forces and civilian Registered Nurses serving in conflict areas and war zones have implications for nurses and nursing practice and identity [47]. The findings implied that nursing leaders, colleagues, families, and friends provide support to these nurses before and after deployment, especially when nurses experienced PTSD. Registered Nurses in the armed forces were required to balance their role responsibilities with those serving armed forces personnel and with that of a Registered Nurse, driven by a mission-focused culture, based on structure and standards [48]. Participants in some of the studies stated that the nursing care provided to all patients in war and conflict areas was of a high standard, thereby creating a sense of increased self-worth and pride as a professional [29,33,37,49]. It was suggested that Registered Nurses who practiced in this context were less likely to be negatively affected psychologically by the presentation of casualties, high activity levels, or the harsh atmosphere [40].

An important finding related to safe nursing practice related to communication and information sharing. This was seen as essential where casualties may be in danger in both war and conflict areas and civilian settings, where communication problems emerge [50]. Therefore, seeking to ensure effective communication and information sharing is a vital component of safe care, and essential in war and conflict areas. From a nursing practice perspective, safe and responsive care in a war and conflict situation centres around collaboration within a multi-disciplinary team [24,27]. Arising from this is the opportunity for Registered Nurses to improve and enhance practice in the future by sharing the lessons learned when caring for injured armed forces personnel and civilians [27,28,33,42,44]. Studies included in this review highlighted the need to improve nursing practice in war and conflict areas by developing clinical skills and wider skillsets thereby aiming to sustain life [25,26,27,38,45].

Although many Registered Nurses appear to thrive in war and conflict areas, as with other areas of the nursing practice, it was vital to be aware of and to respond to the multiple pressures. Stressors such as frequent movements to new assignments, detachment from trusted and familiar support systems, and the possibility of active deployments may lead to additional stress for some Registered Nurses and their families. Some study participants talked about how persisting mental health issues including compassion fatigue and grief over the loss of life in the armed forces contributed to the stress [23,27,29,40]. Caring for children in war and conflict areas was a particular source of anxiety, owing to a lack of clinical experience in this area of practice. Despite these obstacles, the researcher recognised several facilitators, including military-provided anticipatory support, the programs, and exit counselling. The Registered Nurses stressed the importance of talking about the impact of the changes and the need for additional support from family and other veterans [27,31,33,35,42,44,46]. Therefore, recognising and responding to the distinct support needs of Registered Nurses servicing in the armed services is required. This is important as research evidence suggests that work environments that provides Registered Nurses with access to knowledge, equipment, support, and opportunities foster a culture of satisfaction, reducing turnover [51]. There is therefore an opportunity to recognise the specific needs to new Registered Nurses to the armed forces and to provide support that enables them to acclimatise to the different culture and expectations. For this to be effective, Registered Nurse leaders should encourage the free flow of information and ideas at all levels through formal and informal processes, as well as maximize possibilities for professional autonomy [52]. Annual seminars, conferences, workshops, and ongoing in-service education are examples of how this can be enabled. Lastly, in relation to the recommendations of the papers, 16 studies recommended for further research in studied area [23,25,26,27,28,29,30,31,35,40,41,42,44,46], and 8 studies recommended that data should be collected to support nursing practice, support systems before and after going deployment, and interventional approaches needed for PTSD [24,33,36,44,45].

## 6. Conclusions and Implications for Practice

In conclusion, civilian and Registered Nurses in the armed forces reported struggling and being challenged emotionally, psychological and physically pre- and post-deployment to war zones and conflict areas. Medical services are one of the most important prerequisites for a successful operation. Health personnel in the armed forces, often Registered Nurses, play a critical role in prehospital care and have a significant impact on wounded troops’ overall survival and mortality rates. Separating the job of armed forces healthcare workers providing medical support system from that of those who are deployed on the frontlines and deliver nursing care to personnel in combat is critical. Based on the included studies, this review identified the role of Registered Nurses deployed in the armed forces and the obstacles to their successful implementation, such as a lack of information at the decision-making level, the necessity for psychological supports, and role-appropriate medical readiness training. Medical readiness training is advantageous to nurses assigned, but it requires advanced knowledge and practice owing to the constantly changing environment in war and conflict areas. Furthermore, while nursing works to substantiate and sustain advanced practice, excluding nurses from the advanced position might have a substantial impact on the care of those injured patients.

A potential armed forces member must be qualified and competent to contribute in an operational capacity. Registered Nurses who are service-ready will prosper in the armed forces. The experiences of military or civilian professional nurses should be used to assess the pre-deployment preparation needs of these nurses and the elements that demand more attention during the preparations phase where they are required to serve in neighbouring countries. Challenges need to be overcome, with a need for proactive planning to address them. To make effective use of Registered Nurses, it is necessary to provide pre-deployment preparation programmes to ensure they are better informed and adequately equipped, with a clear understanding of role expectations, and to ensure there is access to adequate support, both psychologically and physically before and after deployment to war and conflict areas.

### Limitations and Strengths of the Study

This study inherits a limitation of not refocusing only on including qualitative studies and conducting a meta-synthesis. However, the choice of including qualitative, quantitative and mixed-methods studies provides an overall picture of military and civilian Registered Nurses’ views on and experiences with delivering care in war and conflict areas.

## Figures and Tables

**Figure 1 healthcare-10-02168-f001:**
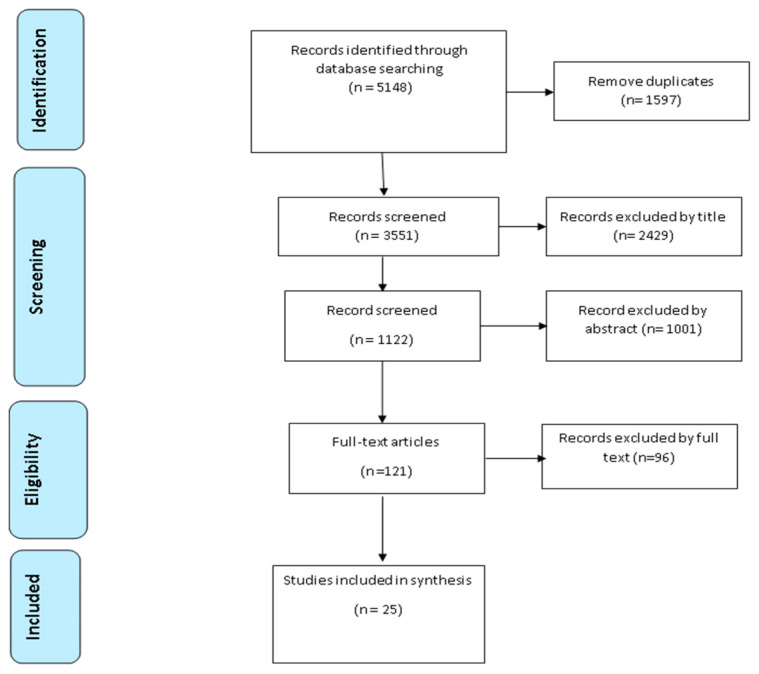
PRISMA flow chart of the systematic review.

**Table 1 healthcare-10-02168-t001:** Studies included in the systematic review (*n* = 25).

No.	Study Citation and Country	Aims	Sample	Methods	Key Findings	Recommendations
1	Agazio (2010) United States of America (USA)	To present a detailed overview of Army nursing practice problems in military operations other than war (MOOTW) compared to recent wartime operations such as Operation Enduring Freedom and Operation Iraqi Freedom.	75 army nurses	Qualitative Focus group interview	This study’s main findings were that nurses require competencies in wartime and MOOTW; patient care demands—deployed environment, getting down to basics, different level of care and training.	Further research is required for a more comprehensive survey to validate the skills and competencies found by this survey.
2	Finnegan et al. (2015) United Kingdom	To investigate the attributes and principles that military nurses possess when performing operational duties in war zones.	18 British army nurses	Semi-structured interviews	The qualities required of military nurses War Zone nursing identified and the practical personal developments required to use their strengths	Further research required to identify the transferability to other Armed Forces and civilian practice internationally.
3	Biedermann (2001) Australia	To identify the essence of nursing’s role in the Vietnam War and increase the raise knowledge and understanding of nurse veterans’ experiences.	17 Australian army nurses	Qualitative interviews	The majority of nursing sisters sent to Vietnam between 1968 and 1971 had no prior knowledge of the nature of work or environment and were clinically unprepared. Nurses were able to adapt professionally in the war zone; their memories of their experiences had a personal impact on their lives.	Further historical research is required to identify the nursing role and experiences during the Vietnam War.
4	Biedermann & Harvey (2001) Australia	To explore the Australian Army nurses’ experiences who were involved in the triage and resuscitation of critically wounded allied and enemy soldiers in the Vietnam War between 1967 and 1971.	17 Australian army nurses	Qualitative interviews	What are the findings?	Further research is required to understand the preparation and support needs of Army Nurses in triaging and resuscitation of critically wounded soldiers.
5	Ekfeldt (2015) Sweden	To detail military nurses’ experiences of preparations for military action in a war zone.	Seven male and one female army nurses	Qualitative structured interviews	Main themes identified were: 1-Recognizing challenges, preparing for the transition from civilian care, preparation for work in a complex context and preparing to deal with anxiety 2-Making informed choices, preparation by investigating one’s motives, preparing by investigating one’s professionalism and preparation through insight into the unique meaning of the caring relationship	Further research is required to more fully understand the preparation needs of Army Nurses in a combat environment and enable reflection skills.
6	Conard & Scott-Tilley (2015) United States of America (USA)	To identify female nursing veterans’ experiences of war on their physical and mental health.	12 female nursing veterans	Qualitative Face-to-Face Interviews	Seven themes identified: living in constant fear while deployed, combat has different meanings, bringing the war home, fear of being forever changed, disrespect from fellow military members, physical health—for better or worse, and combat has rewarding experiences.	Future research needed to identify the first-hand accounts of female Gulf War II nursing veterans from all military areas regarding their experience of female officers and enlisted personnel in Gulf War II.
7	Conlon et al. 2019 Australia	To understand the lived experiences of nursing officers when working as a member of a military trauma team.	6 nursing military officers	Qualitative Interviews	The findings included telling their stories; the role—who we are and what we do; the environment—it is so different; training—will it ever fully prepare you; working in teams—there is no “I” in the team; and leadership—will the real leader please stand up.	No specific recommendations for future research were made.
8	Kenny & Hull (2008) USA and Europe	To examine the stressors of nurses working in the ICUs of two U.S. military medical treatment facilities (MTFs) before and after the beginning of the wars in Iraq (Operation Iraqi Freedom [OIF]) and Afghanistan (Operation Enduring Freedom [OEF]).	10 ICU nurses	Survey & Questionnaire	The finding highlight that that the “stay-behind” nurses at both MTFs have experienced significant stress from (1) the deployment of colleagues and, in some cases, family members; (2) changing missions related to the care of wounded soldiers returning to the medical centre; and (3) the extra work created by deployments, with few backfill replacements, and the influx of younger patients who have increased levels of acuity and need.	Further research to identify the impact of compassion fatigue and resiliency required in military MTFs settings and the precautionary and preventative approaches required.
9	Elliott (2015) United States of America (USA)	To describe the military nurses’ experiences during the post-deployment phase and describe the post-deployment experience’s meaning.	10 military nurses	Interviews	Five themes were identified from the data: learning to manage changes in the environment; facing the reality of multiple losses; feeling like it’s all so trivial now; figuring out where I ‘fit’ in all the chaos; and working through the guilt to move forward. The meaning of the participants experiences included serving a greater purpose and looking at life through a new lens.	Further research is indicated to identify the benefits seeking and making sense in relation to military nurses’ experiences could provide additional support for interventions at different stages of the deployment period.
10	Finnegan et al. (2016) Afghanistan	To explore the challenges and psychological stressors facing military nurses in undertaking their operation	18 military nurses	Interviews	The factors that caused stress were identified during both deployment and returning home, and measures to address the need for rest and exercise due to being deployed in a war zone.	There is a need for further research to identify the possible stressors that can impact on military nurses in their return home.
11	Firouzkouhi et al. (2013) Iran and Iraq	To identify the experiences gained by civilian nurses’ activity and through the duration of the Iran and Iraq war.	15 civilian nurses	Interviews	Health care assistants and nurses who joined the armed forces, did not have prior training in this area of nursing, yet many quickly adapted and performed their nursing role successfully.	Studies are required to identify how civilian nurses’ experiences can be used in peacetime to help prepare future nurses with knowledge of wartime nursing expectations.
12	Goodman et al. (2013) Iraq	To understand military nurses’ experiences of care for Iraqi patients.	15 army nurses	Focus Group Interview	The data identified expanding practice, ethical dilemmas, and the cultural divide with details of the opportunities for learning additional knowledge and skills to increase their competence. Nurses experienced mental distress when confronted with ethical issues regarding safe patient care. Lack of trust in interpreters and animosity. Some The nurses’ challenged culturally when caring for a population with a different language, value system, customs, and traditions.	Further research is required to more fully understand the specific challenges experienced by military nurses and support their personal and professional development.
13	Griffiths & Jasper (2008) United Kingdom	To explore the nature of military nursing in an environment of war.	24 military nurses	Focus group Interviews	The findings included the realities of practicing in the military in a war environment and the dual role as a nurse and member of the military and the impact on caring during war and the transition to a ‘warrior’.	Further research is needed to capture the caring nature of the nurses role within a conflict zone to prepare military, NGO, and civilian nurses for their roles.
14	Hagerty et al. (2011) United States of America	To explore the lived experience of combat-wounded patients and the military nurses who provided.	20 military nurses and eight combat wounded-patients	Focus Group Interviews	The findings suggest that nurses and patients have shared experience upon which they can attach meaning and recognize that the experiences led to the ‘changed self’ and the professional and role boundaries that existed.	Further research should focus on identifying strategies to help nurses and patients cope and adapt to the stressful combat circumstances.
15	Han (2019) Korea	To explore the essence of the experience of Korean female nursing officers during the Vietnam War	14 Korean female nursing officers	Interview	The findings identified that some participants experience on-going role confusion while being committed to their nursing role. A deep sense of comradery was experienced within the context of the dark side of war, with a fear of gender discrimination experienced by some, while finding the role rewarding that enabled development.	Future studies should focus on the experiences of nurses as women and officers in the conditions of war to inform future support needs prior to deployment.
16	Lal & Spence (2016) New Zealand	To identify the lived experience of New Zealand nurses providing humanitarian aid within surgical settings in developing countries.	4 nurses	Interviews	For humanitarian work, specialized skills and nursing experience are required yet participants required further preparation, including cultural issues and needs	Future research should focus on contextual and cultural issues that affect nursing in humanitarian settings.
17	Peyrovi et al. (2015) Iran	To analyse the history of the wartime experience of Iranian nurses in the Iran-Iraq war.	13 military nurses	Interview	Finally, data analysis of significant statements from 17 interviews yielded 5 themes and 18 subthemes. (1) “From the margins to the middle” was one of the five themes that emerged. (2) “Referral care growth,” (3) “Personal and professional development,” which includes both personal and professional growth. (4) “War nursing’s emerging cornerstone of society,” and (5) “Threats to nursing at war.”	It Is recommended that the wartime memories of Iranian nurses be collected, archived, and analysed.
18	Rahimaghaee et al. (2016) Iran	To explain nurses’ experiences and views on the care of injured soldiers during the Iraq–Iran war (1980–1988).	14 nurses	Semi-structured Interview	The data revealed two main themes (Care in the war, a different culture and concept, and Care accomplishments during the war) as well as six subthemes (Unusual working conditions, Different work spirit, A real yet informal classroom, Professional self-achievements, Professional community outcomes, The changed self).	No specific recommendations for future research were made.
19	Rivers et al. (2013) USA	To understand the lived experience of U.S. Army nurses’ reintegration and homecoming following deployment to Iraq and/or Afghanistan	22 army nurses	Qualitative Interview	Five themes are identified: (1) aspects of command support were described as “No one cares”; (2) meeting requirements for attendance at pre/post-deployment briefings was described as “Check the blocks”; (3) readjustments from focusing solely on-duty requirements versus multitasking, such as family obligations and everyday life, resulted in the “Stress of being home.” (4) nurses stated “They do not understand” when referring to anyone without deployment experience (family, friends, other soldiers); and (5) when referencing deployment experiences, nurses emphasized that, “It just changes you.”	Future research is indicated that focus on meeting the needs of nurses, such as processes and education to support during deployment or their return to home and family.
20	Rushton et al. (2008) USA	To gather experiences from nurses who have served their country in wartime, either on the front lines or in supportive roles.	10 navy nurse corps and one air force nurse corps	Qualitative Interview	The themes identified: “It is what we’re here for,” demonstrates a commitment to care and to sacrifice. Other major themes drawn from the study included lessons learned from their wartime nursing experiences, sacrifices made, and chronicles of caring.	Further research is required to explore nurse experiences during the war. nurses. It is recommended that families and relative are physically and emotionally prepared in order to provide support to deployed nurse.
21	Scannell-Desch (1996) USA	To explore common components of the lived experience of women military nurses who served and after returning from Vietnam.	24 military women nurses	Qualitative In-depth Interview	The study’s findings revealed that the Vietnam war affects the lives of nurses during and after the war.	Further studies is indicated on Vietnamese women nurses and other classes of people who have worked in war and disaster situations.
22	Scannell-Desch (2005) USA	To describe guidance for nurses today from the lessons learned by nurses who served in the Vietnam War.	24 nurses—nine army, eight navy and seven air force	Qualitative Interview	Seven themes were identified: advice about journaling, training, caring for yourself, use of support systems, talking about your experiences, understanding the mission, and lack of preparation for war.	It is recommended that the facets of war should be included in military nurse training.
23	Scannell-Desch & Doherty (2010) USA	To describe the lived experience of U.S. military nurses who served in Iraq or Afghanistan during the war years 2003 to 2009, and life after returning from war	37 nurses—18 army, four navy, and 15 air force	Qualitative Interview	Seven themes were identified (Deploying to War, Remembrance of War: Most Chaotic Scene, Nurses in Harm’s Way: More Than I Bargained For, Kinship and Bonding: My Military Family, My Wartime Stress: I am a Different Person Now, Professional Growth: Expanding my Skills, Listen to Me: Advice to Deploying Nurses and seven subthemes emerged from data analysis of significant statements. These themes and corresponding subthemes captured the essence of the lived experience of military nurses serving in Iraq and Afghanistan during 2003 to 2009.	The study recommended to adapt the approach in treating nurses with PTSD. Future research is identified to look at war nurses’ perspectives from countries other than the United States.
24	Shamia et al. (2015) Palestine	To establish the association between war traumatic experiences, posttraumatic stress disorder (PTSD) symptoms and posttraumatic growth among nurses in the Gaza Strip, 2 years after an incursion on Gaza and during a period of ongoing trauma exposure.	274 nurses	Quantitative Survey Questionnaire	The study shows 19.7% of the nurses had absolute PTSD. There was a strong link between traumatic events and PTSD ratings and traumatic events in the community and posttraumatic development.	It is recommended nurses with PTSD should receive mental health assistance that can provide instruction, counselling, and assistance to other healthcare providers.
25	Tjoflåt (1997) Norway	To describe the nursing practice in war zones and skills the nurses have acquired from work	39 nurses	Mixed-Method Questionnaire	The findings suggest that the nurses were well-prepared for this mission. The nurses gained personal and technical skills while serving in war zones.	Further, qualitative research is needed to learn more about the nursing profession in war. A study like this may help researchers better understand patients, nurses, and the unique practice situation.

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
