# Peer review of "Registered Nurses’ Views and Experiences of Delivering Care in War and Conflict Areas: A Systematic Review"

_healthcare, 2022, doi:10.3390/healthcare10112168_

Round 1

Reviewer 1 Report

Dear Authors, the manuscript “The Views and Experiences of Registered Nurses of Delivering Care in War and Conflict Areas: A Systematic Review of the International Research Evidence”, treats an interesting and innovative topic for international readers. It treats an interesting issue however, I have some suggestions to improve your manuscript.

Title: Why do you highlight in the title International literature? Since it is a systematic review it is a granted, I suggest removing.

Background, results and discussion are very interesting and well detailed.

The major concern is regarding the methods. Why didn't you select only the qualitative studies (being almost all of them) and conduct a meta-synthesis?

The review you conducted is well described and methodologically sound and can be published that way. In my opinion focusing only on the qualitative studies with a meta-synthesis could be even more innovative. It is just a suggestion and I understand this means revising a lot of the paper. You could then report a reflection on this in the limitations or strengths section depending on how you want to describe the choice made.

Author Response

Manuscript ID: healthcare-1987264
Type of manuscript: Review
Title: The Views and Experiences of Registered Nurses of Delivering Care in War and Conflict Areas: A Systematic Review of the International Research Evidence

Revised Title: The Views and Experiences of Registered Nurses of Delivering Care in War and Conflict Areas: A Systematic Review

Authors’ Reply to Reviewer 1 Comments

COMMENT 1: Title: Why do you highlight in the title International literature? Since it is a systematic review it is a granted, I suggest removing.

RESPONSE: Thank you for this comment. The phrase “of the International Research Evidence,” has been removed as suggested.

COMMENT 2: Background, results and discussion are very interesting and well detailed.

RESPONSE: Thank you very much for this comment.

COMMENT 3: The major concern is regarding the methods. Why didn't you select only the qualitative studies (being almost all of them) and conduct a meta-synthesis?

RESPONSE: Thank you for this comment. Our systematic search resulted in qualitative, quantitative and mixed-method studies. Deciding to exclude the quantitative and mixed-method articles would undermine the overall picture of our study findings.

COMMENT 4: The review you conducted is well described and methodologically sound and can be published that way. In my opinion focusing only on the qualitative studies with a meta-synthesis could be even more innovative. It is just a suggestion and I understand this means revising a lot of the paper. You could then report a reflection on this in the limitations or strengths section depending on how you want to describe the choice made.

RESPONSE: Thank you for this comment. We provided a Limitations and Strengths of the Study section before the Author Contributions section.

Reviewer 2 Report

The literary review is composed magistrally, following all the steps and rules of this kind of research. Methods and exclusion/inclusion criterias are thoroughly delineated. The narrative synthesis of the results flows perfectly and it's easy to read. 

As a reader, I am unconvinced by the use of a table to present the description of each study. I think the work of synthetizing and criticizing findings could be more interesting than the introduction of summaries of the single works appraised. 

More emphasise should be placed on a critical appraisal of studies and the methods used. Although this is not a systematic literary review, it could be interesting to have a better overview of the challenges that researchers might have encountered in conducting qualitative and quantitative research on the topic.

The article seems valuable, above all in current times and provides a good insight on the researches that have already been done. 

I reccomend the publication of the article, with minor changes.

Author Response

Manuscript ID: healthcare-1987264
Type of manuscript: Review
Title: The Views and Experiences of Registered Nurses of Delivering Care in War and Conflict Areas: A Systematic Review of the International Research Evidence

Revised Title: The Views and Experiences of Registered Nurses of Delivering Care in War and Conflict Areas: A Systematic Review

Authors’ Reply to Reviewer 2 Comments

COMMENT 1: As a reader, I am unconvinced by the use of a table to present the description of each study. I think the work of synthetizing and criticizing findings could be more interesting than the introduction of summaries of the single works appraised. 

RESPONSE: Thank you for this comment. The table serves and gives summary for the readers and we would like to request from the honorable reviewer to retain the table.

COMMENT 2: More emphasise should be placed on a critical appraisal of studies and the methods used. Although this is not a systematic literary review, it could be interesting to have a better overview of the challenges that researchers might have encountered in conducting qualitative and quantitative research on the topic.

RESPONSE: Thank you for these comments. We already addressed these parts in Section 4.2. Characteristics of Included Studies. We would like to clarify if the honorable reviewer needs additional information.

COMMENT 3: The article seems valuable, above all in current times and provides a good insight on the researches that have already been done. I reccomend the publication of the article, with minor changes.

RESPONSE: Thank you very much for these comments.

Reviewer 3 Report

Please see the comments

Author Response

Manuscript ID: healthcare-1987264
Type of manuscript: Review
Title: The Views and Experiences of Registered Nurses of Delivering Care in War and Conflict Areas: A Systematic Review of the International Research Evidence

Revised Title: The Views and Experiences of Registered Nurses of Delivering Care in War and Conflict Areas: A Systematic Review

Authors’ Reply to Reviewer 3 Comments

COMMENT 1: The authors state that they scanned 4 databases for the papers. For less informed readers, the authors should explain why these 4 databases are chosen and why leaving other databases would not leave out important related studies. 

RESPONSE: Thank you for this comment. We added an explanation of choosing the four databases in Section 3.1. Search Strategy.

COMMENT 2: The authors used PRISMA to eliminate papers. Again, the authors can briefly explain (maybe even in a footnote) why they used this method. For example, is this something standard in systematic reviews? Are there other methods that other people use in systematic reviews and if so why PRISMA etc.

RESPONSE: Thank you for these comments. We added details/reason on why we used PRISMA in Section 3.3. Study Screening, stating that: “The PRISMA 2020 statement guided and enabled the researchers to report this systematic review that allows the readership to evaluate the appropriateness of the methods used, leading to the trustworthiness of the review findings.”